# A Theocentric Environmental Ethic

**Garrett J. DeWeese**

Department of Philosophy, Talbot School of Theology, Biola University, La Mirada, CA 90639, USA; deweese@biola.edu

**Abstract:** An influential view among environmentalists and ecologists is that religion, in general, and Christianity, in particular, not only have nothing to offer to environmental ethics but are actually hostile to the environment. I argue that a biblically informed theocentric environmental ethic of stewardship offers rich resources for duty-based environmental ethics in general and, in particular, for establishing grounds for restoration, conservation, and preservation of the environment.

**Keywords:** environmental ethics; Genesis; dominion theology; stewardship; conservation; preservation; restoration





## 1. Introduction: Historical Perspective

In 1967, Lynn White, professor of medieval history at UCLA, delivered an influential address to the American Association for the Advancement of Science, later published in the journal *Science* with the title "The Historical Roots of our Ecological Crisis."[1] According to White, the Western Judeo-Christian tradition in general and the Bible in particular bear "a huge burden of guilt" for the contemporary environmental crisis.

Six years later, in 1973, the eminent historian Arnold Toynbee, in an essay entitled "The Genesis of Pollution," made a similar claim:

> The thesis of this essay . . . is that some of the major maladies of the present-day world—in particular the recklessly extravagant consumption of nature's irreplaceable treasures and the pollution of those of them that man has not already devoured—can be traced back to a religious cause, and this cause is the rise of monotheism. (Toynbee 1973)

Noted landscape architect Ian McHarg echoes this viewpoint: "In its insistence upon dominion and subjugation of nature, [the biblical creation story] encourages the most exploitative and destructive instincts in man rather than those that are deferential and creative." (McHarg 1969; cited by Wright 1989). Feminist theologian Rosemary Radford Reuther maintains that the Christian tradition, as heir of Neoplatonic dualism, presupposes a thoroughgoing alienation of humanity from nature. Consequently, she assigns Christianity considerable blame for "this debased view of nature, as the religious sanction for modern technological exploitation of the earth." (Reuther 1972). Examples of this view of the Judeo-Christian tradition as perpetrator of, or at least willing accessory to, the rape of nature, could be multiplied. So, it is no surprise that since the birth of the environmental movement some six decades ago,[2] most attempts to offer a systematic ethical framework for human interaction with nature, presupposing as they do something like White's or Toynbee's thesis, have seemed to many Jews and Christians to be deeply flawed.

I will argue that a theocentric ethic of environmental stewardship is able to answer criticisms such as White's and Toynbee's by offering a constructive way to make sense of values in nature, and also to speak to the central issues in environmental ethics: preservation, conservation or sustainability, and—perhaps more problematic—restoration.[3] I will also briefly argue that a theocentric ethic overcomes deficiencies in each of the four most popular contemporary approaches to environmental ethics.

## 2. Contemporary Perspectives: A Brief Summary

To simplify somewhat, there are four contemporary approaches to environmental ethics that are widely influential but, I believe, are also seriously deficient. Briefly summarizing these approaches will serve to put in sharper relief the theocentric stewardship ethic that I will present. The four approaches are (1) *biocentric*, (2) *ecocentric*, (3) *"deep ecology"*, and (4) *minimalist* environmental ethics.

### 2.1. A Minimalist ethic

I will begin with this last approach, a *minimalist* environmental ethic. Those holding to this view assert that we simply have no moral responsibility for how we use—or abuse—nature itself; our moral relations with the natural world are conceived in purely anthropocentric terms.

John Stuart Mill, for example, was skeptical about an ethic of nature:

> In sober truth, nearly all the things which men are hanged or imprisoned for doing to one another, are nature's everyday performances . . . . Nature impales men, breaks them as if on the wheel, casts them to be devoured by wild beasts, burns them to death, crushes them with stones like the first Christian martyr, starves them with hunger, freezes them with cold, poisons them by the quick or slow venom of her exhalations, and has hundreds of other hideous deaths in reserve . . . . Everything, in short, which the worst men commit either against life or property, is perpetrated on a larger scale by natural agents. (Mill 1874)

William James uses even stronger language:

> Visible nature is all plasticity and indifference,—a moral multiverse . . . and not a moral universe. *To such a harlot we owe no allegiance*; with her as a whole we can establish no moral communion; and we are free in our dealing with her several parts to obey or to destroy, and *to follow no law but that of prudence* in coming to terms with such of her particular features as will help us to our private ends.[4]

The views of Mill and James are reflected in contemporary indifference to environmental concerns such as air and water pollution and even hostility towards the whole notion of climate change. As the evidence for the anthropogenic nature of climate change grows, such indifference to environmental concerns seems misguided.

Some Christians also embrace a minimalist environmental ethic, apparently believing that God will never allow his world to be destroyed by humans fulfilling the mandate to "subdue and rule" the earth.[5] They believe that acknowledging any moral duty with respect to creation must, in a zero-sum game, inevitably detract from more salient biblical commands such as evangelism. However, just because the care of creation is not the primary duty of Christians, it does not follow that it is not a duty at all. I will not say more about the rejection of environmentalism by some Christians; I believe the positive case for stewardship of creation is sufficient to rebut the rejection of environmental concern. I do want to note before moving on, though, that those most directly affected by any environmental degradation are more often than not the poorest of the earth's peoples, and that should deeply concern Christians.

### 2.2. A Biocentric Ethic

The other three approaches seem misguided to me. A *biocentric* environmental ethic grounds values and duties to the environment in a generalized respect for life or nature.[6] Biocentrism stands on certain principles: (1) Humans are members of Earth's Community of Life on the same terms as other species. (2) All species, including humans, are integral elements in a system of interdependence. (3) Each organism has its own good, which it pursues in its own way. Finally, (4) Humans are not inherently superior to other species.

Unfortunately, for Christians, biocentric approaches suffer from an impoverished anthropology, failing to recognize that God's image in every man, woman and child sets humanity off from the rest of nature in a very important way. However, this approach

is problematic for secularists as well. For the very abilities that enable humans to think inductively, to use language to express causal relations and not merely correlations, and above all, to make moral judgments about the value of life—those abilities do indeed confer on humans a status superior to other life. In other words, the biocentric approach asks humans to use the capacity of moral judgment unique to humans to deny that humans are morally unique.

### 2.3. An Ecocentric Ethic

An *ecocentric* environmental ethic grounds values and duties in the natural world itself, for example, ecosystemic homeostasis—a sustainable balance between plant and animal species in ecosystems.[7] Holmes Rolston III, a Christian scholar often regarded as the "father of environmental ethics," argues cogently that the natural world has value just because it is "value-able", that is, has value in itself in some way not necessarily tied to human valuers, and this avoids the so-called "naturalistic fallacy", the attempt to derive a moral "ought" from a non-moral "is". Rolston writes, "Tourists in Yosemite do not value the sequoias as timber, but as natural classics, for their age, strength, size, beauty, resilience and majesty. This viewing constitutes the trees' value." (Rolston n.d.). But it seems that an ecocentric approach generally ignores wild fluctuations in the Earth's natural history, fluctuations which would seem to erase previous values. How ecosystemic homeostasis is then a value remains unclear. And the adaptive ability of organisms evidenced when ecosystems change would seem to demand fluctuating values, and it is not clear how such unstable values ground environmental concerns.

Further, it also ignores the many purposes for which God can *use* his creation. If maintaining balance in ecosystems is a moral obligation, then the many biblical references to God's use of nature in blessing or in judgment on humans would be quite problematic. An ecocentric ethic thus places creaturely values above the Creator's, and so is problematic for Christians.

### 2.4. A "Deep Ecology" Ethic

*"Deep ecology"* and "eco-feminism" are closely associated with a resacralizing approach that draws on Native American, Wiccan and Eastern traditions to ascribe some sort of sacredness primarily to non-human species and landscapes.[8] Resacralization is often grounded in James Lovelock and Lynn Margulis's Gaia hypothesis, in which all the Earth's living matter, air, oceans, and land surface form a complex, self-regulating and self-directing system *that can be seen as a single organism.*[9] Gaia, this single vast system itself, is deified, and "Mother Earth" becomes "Mother Goddess Earth," more than a metaphor. Relying less on quasi-religious concepts, some "deep ecology" advocates argue that "Massive human diebacks would be good. It is our duty to cause them. It is our species' duty, relative to the whole, to eliminate ninety percent of our numbers." (Aiken and Regan 1984).

The shortcomings of this approach should be clearly apparent and not just to Christians. If the "elimination" of ninety percent of humanity involves genocide, then it is calling for a gross violation of human rights. And if it is calling simply for a natural reduction of the population, it is up to the advocate of the position to show how this can be achieved without vast human suffering and rights violations.

Much more could be said, of course, about all four of these approaches to environmental ethics, but this is not the place. The popular approaches either devalue nature or denigrate humanity. I will now argue that an ethic of creation care that is theocentric at its core is able to avoid these deficiencies and offer a constructive way to make sense of values in nature.

## 3. Stewardship as the Theocentric Ethical Category

Naturally, the arguments of White and Toynbee did not go unchallenged. Beginning perhaps with the publication of Francis Schaeffer's book, *Pollution and the Death of Man: The Christian View of Ecology* in 1970 (Schaeffer 1970), many Christians challenged the

hostile view, developing differing theological and ethical grounds for "creation care." A salient theme stressed the biblical image of *stewardship* as the proper Christian attitude toward nature.[10] Perhaps because White and successive critics harshly denounced the "dominion theology" of Genesis 1:28 (which reads, in the language of the King James Version, "Be fruitful, and multiply, and subdue the earth, and have dominion over it"), Christian response has focused on the more positive biblical theme of stewardship.

James Nash, for example, writes,

> According to one popular conception—actually, a misconception and stereotype—of "dominion," humankind is a distinctive creation designed for dominion . . . Nature is simply matter, resources waiting to be reformed for human utility. This viewpoint embodies the fundamental failures at the roots of the ecological crisis . . . Without doubt, Christian traditions bear some responsibility for propagating these failed perspectives. Consequently, the ecological crisis is a challenge to Christians to eradicate the last vestiges of these ecologically ruinous myths. (Nash 1993, p. 19)

But Nash, one of the few to carefully analyze the history of the ecological movement, concludes,

> Thus the ecological complaint against Christianity appears to be a serious historical oversimplification . . . [Dominion] became isolated from the moderating and controlling influences of the whole corpus of Christian thought and served as a license for elimination with extreme prejudice. The practices under the rubric of dominion were alien to the biblical and most traditional understandings of the concept. (Nash 1993, pp. 77, 79)

In surveying the growing literature on creation care coming from Christian writers, it appears to me that, with only a few exceptions, Christians have tended to avoid the dominion passages, and so their presentations of biblically-based environmental ethics are somewhat unbalanced. Certainly, stewardship is a more prevalent image in the Bible than subduing or having dominion. Yet if we are to do justice to a theocentric environmental ethic, we must take into account both the positive connotations of "stewardship" and the negative connotations of "dominion," since both are aspects of the biblical witness.[11]

In the following section, I will unpack the notion of God as "owner" and humans as "stewards" of creation and explore the category of stewardship as it applies to our relationship with God's creation. Next, I will pay attention to the alleged negative connotations of subduing and having dominion looking at three specific environmental duties of stewardship: restoration, conservation, and preservation. Stewardship is a deontological category and stresses the steward's duty to the sovereign. I have chosen to develop this account in deontological terms to contrast to what, in my view, is a misguided attempt to explicate duties to species, ecosystems, future generations, and such. In my view these are not the sorts of things to which we can have duties. As will be shown, a theocentric ethic can make sense of duties to God *with respect to* such things.[12]

### 3.1. Stewardship and Ownership

The biblical record is clear. "In the beginning God created the heavens and the earth" (Genesis 1:1), and the act of creation implies the right of ownership: "The earth is the Lord's, and everything in it" (Psalm 24:1). God as creator—and therefore owner and master of creation—is a salient concept throughout the Bible. Many implications flow from this fundamental affirmation of God as creator and master of all; two in particular are relevant here. First, if God is Creator, then he must impart something of himself to his creation.[13] In Psalm 19:1, David sings, "The heavens declare the glory of God; the skies proclaim the work of his hands," and the Psalmist can call on creation to praise its Creator:

> Praise the Lord from the earth, you great sea creatures and all ocean depths,
>> lightning and hail, snow and clouds, stormy winds that do his bidding,
> you mountains and all hills, fruit trees and all cedars,

> wild animals and all cattle, small creatures and flying birds,
>
> kings of the earth and all nations, you princes and all rulers on earth
>
> young men and maidens, old men and children. (Psalm 148:7–12)

The writers of the Psalms imbue their songs with nature parables illustrating God's goodness, beauty and faithfulness. Jesus used the birds and flowers to illustrate God's goodness (Matthew 6:27–30). The Apostle Paul as well maintains there is revelation of the character of God in creation (Romans 1:20). Nature is God's handiwork, and in it, we can see reflected something of his character, even though finitely and imperfectly.

Second, as his possession, nature is God's to employ as he pleases. Psalm 78 records God's use of nature for his purposes:

> He divided the sea and led them through; . . .
>
> He guided them with the cloud by day and with light from the fire all night.
>
> He split the rocks in the desert and gave them water as abundant as the seas; . . .
>
> He gave a command to the skies above and opened the doors of the heavens;
>
> He rained down manna for the people to eat, he gave them the grain of heaven . . . .
>
> He rained meat down on them like dust, flying birds like sand on the seashore. . . .
>
> Again and again they put God to the test; they vexed the Holy One of Israel. . . .
>
> He sent swarms of flies that devoured them and frogs that devastated them.
>
> He gave their crops to the grasshopper, their produce to the locust.
>
> He destroyed their vines with hail, their sycamore-figs with sleet . . . . (Psalm 78 *passim*)

The biblical perspective is that just as nature was created by God and can reveal his character, so also nature belongs to God and can reveal his purposes for blessing or punishment. This perspective alone could well be sufficient to ground an ethic of respect for nature, just as understanding who is the owner of new construction and what his purposes in the project could (one hopes) be sufficient to prevent vandalism, pilfering and graffiti at the project site.

Thus, as we revel in extravagant displays of wildflowers in the mountains of the High Sierra or thrill to the bugling of elk in the Rockies, as we smile at the anhinga hanging its wings out to dry in the Everglades or delight in a sunset behind the haystack rocks of Rialto Beach, as we watch in wonder the elephant seals at Piedras Blancas or the lurking alligators in Okefenokee Swamp, we will express gratitude to the Owner of this planet for allowing us to enjoy what he has made and to learn what we can from his creation, and we resolve to the best of our ability to refrain from marring his property. We feel privileged to be a part of the same creation as the delicate blossoms of the columbine or the awesome humpback whale, and the experience of the wild arouses within us a feeling of something like kinship with nature.

However, in recognizing we are guests on another's property, albeit tacitly, we acknowledge that the owner is the sovereign lord of their land. And so it comes as a sobering realization, heavy with responsibility, to find that we are designated the caretakers, or stewards, of this vast and varied world.

Just here we must draw clear and distinct lines of demarcation between the theocentric concept of humans as stewards of creation and the deficient concepts of humanity contained in biocentric, ecocentric, or deep ecology environmental ethics. While a theocentric ethic takes seriously the conception of humans as one with the environment in terms of their nature as creatures, it further recognizes that these creatures are endowed by the Creator with the capacity to bear a special delegated responsibility. In regards the transcendent infinity of the Creator, humans stand on the same side of a gulf as the humblest earthworm. We are *in and with* nature. However, in regards the immanent personality and moral agency of the Creator, humans stand on his side of the gulf, over against the rest of creation. For as bearers of the image of God, as being created with the purpose of rulership (more on which

below), humans are in a relevant sense *over* nature. This may sound like speciesism to some, but it is perhaps the only view of human beings that does justice both to our nature as rational, moral agents capable of seriously affecting our environment and responsible for those effects, and at the same time as fellow-creatures of the biosphere with all other life. And it seems that virtually everyone, either explicitly or implicitly, recognizes this in both their thinking and their acting. The concept of humans as stewards of creation, and hence over creation, is not an anthropocentric but a theocentric concept. This delightful, wonderful, extravagantly furnished world reveals God's character and serves his purposes, and he has placed us in it and made us stewards over it.

*3.2. Stewardship and Responsibility*

The English word "steward" entered the language sometime in the eleventh century as

*stigwaerd*, *stig* probably referring to a house or some part of a house or building, and *waerd* [later, *ward*] meaning of course "warden" or "keeper." The first meaning offered by the *Oxford English Dictionary* is this: "An official who controls the domestic affairs of a household, supervising the service of his master's table, directing the domestics and regulating household expenditures; a major-domo." (Hall 1990, p. 40)

The most common use of 'steward' in English Bibles is to translate the Greek word *oikonomos*, and that of 'stewardship' is to translate *oikonomia*. These words are composed of *oikos*, "house, household," and *nomos*, "rule, law." So it follows that the steward is one charged with the rule of the household, responsible for the great natural wealth of the master's estate.

And there is still more to the conceptual reach of "steward." A century ago, the biologist Ernst Haeckel coined the word *ecology* to mean

[T]he knowledge of the sum of the relations of organisms to the surrounding outer world, to organic and inorganic conditions of existence; *the so-called "economy of nature,"* the correlations between all organisms living together in one and the same locality, their adaptation to their surroundings, their modification in the struggle for existence. . . . [14]

Haeckel formed *ecology* from *oikos*, "household," plus *logos*, "study." Thus the relationship between stewardship and ecology is a very close one conceptually.

Clearly, the steward was not charged with responsibility over valueless bric-à-brac, and neither does a stewardship model of environmental ethics regard non-human creation as valueless. Rather the owner or monarch would place the most trusted servant as steward over the most valuable parts of the estate. But that value is derived from its relation to the owner, not to the steward. The steward values *what* the master values, values *just as* the master values, and manages the estate *in the manner* the master would want it managed were they present. That is why the Christian concept of stewardship is, at its core, thoroughly theocentric rather than anthropocentric. Wilkinson writes,

[T]hough the value of things must comport with God's principles for correct valuing, humans are still the Creator's agents for this task. . . . The candidates for such principles put forward by the world around us—purely subjective valuations, usefulness for human purposes, and intrinsic values—are all deficient by Christian standards. The missing element is transcendence: valuing of the creation ought to be grounded in the Creator's norms. (Wilkinson 1990, p. 239)

In this way, a theocentric ethical system solves the problem of valuation that troubles naturalistic ethical systems. The concept of intrinsic value is contested because it seems, at least intuitively, that value in nature can never be more than instrumental.[15] But in a theocentric ethic, all aspects of the natural world have intrinsic value because of who their Creator is, irrespective of their instrumental value for humans or other creatures. Since "The heavens declare the glory of God" (Psalm 19:1), a clear night sky is of value in order that

the glory of the creator might shine regardless of the wishes of observational astronomers, lovelorn poets or aspiring natural theologians. The abiotic lunar landscape has value, even though only twelve humans have ever set foot on the moon, simply because it is a part of creation that God put in place (Genesis 1:16). That God will, in the last days, use it as a "billboard" (Luke 21:25) only adds instrumental value to the moon's antecedent intrinsic value as God's handiwork.

So the Creator determines the value of creation, and the steward is the one in charge of what is valuable. The picture is vice-regency, orderly management, not exploitation. (The concept of vice-regency, familiar to biblical theologians, has recently received more prominence in discussions of the *imago dei*.) The Psalmist, rhapsodizing on the creation, echoes the teaching that rulership is part of the purpose for which God created humanity.

> When I consider your heavens, the work of your fingers,
>> The moon and the stars, which you have set in place,
> what is man that you are mindful of him,
>> the son of man that you care for him?
> You have made him a little lower than God
>> and crowned him with glory and honor.
> You made him ruler over the works of your hands;
>> you put everything under his feet:
> all flocks and herds, and the beasts of the field,
>> the birds of the air, and the fish of the sea,
>> and all that swim the paths of the seas. (Psalm 8:3–8)

Commenting on this psalm, theologian Henri Blocher writes,

> Psalm 8, which sings the paradox of the smallness of man, and the glory of his position, may be read in line with this thought. . . . This royal capacity does not authorize any tyranny: the reign of the creature-image cannot be other than a lieutenancy; man is a vassal prince who follows the directives of the Sovereign and is accountable to him. On the other hand, the king serves as mediator for blessing the land: thus, the man is for the earth; as a shepherd, he dominates the animals with a view to their good as his own.[16]

Blocher's reading is supported by such Old Testament commands as "Do not muzzle an ox while it is treading out the grain" (Deuteronomy 25:4) and "A righteous man cares for the needs of his animal" (Proverbs 12:10). So it is clear that a biblical stewardship ethic grounds environmental responsibility.

### 3.3. Stewardship and Duty

There is one last corollary of the concepts of ownership and stewardship, which is pertinent to environmental ethics. The steward's duty to the owner is to fulfill the owner's mandate, to act in their stead, and to maximize the owner's good in relation to the stewardship. As an analogy, consider the medieval stewardship of a sheriff over a county (the politics of Robin Hood's era, for example). The sheriff was responsible not only for such matters as collecting taxes and conscripting a quota of soldiers for the monarch's army but also for matters such as securing the county from marauders, enforcing justice in the county, and "promoting the general welfare." But notice that these duties are owed by the sheriff to the king; the people of the county have only a "third-party interest" in the duties and do not themselves have rights against the sheriff.

This analogy helps clarify one persistent problem in environmental ethics: the question of whether nonhuman living things possess rights. A number of ethicists and jurists have grave worries about granting rights to nonhuman creatures. (For example, Taylor 1986, p. 219ff). However, from a theocentric—but not from a biocentric or ecocentric—point of view, we see that we *do* owe duties to God *with respect to* our treatment of creation. Thus, in

a theocentric environmental, ethical system, both future generations and even species, the extinction of which is attributable to human action or inaction, might have just claims. But the claim would not be brought by either future generations or by animal species (whatever that would mean), but rather by God himself on behalf of the third party whose welfare was violated by the steward.

## 4. Stewardship as Restoration

Restoration, roughly, is concerned with what moral considerations guide us in determining what and how much humans should do to repair damage caused to natural areas and, further, whether and to what degree we should seek to repair natural areas that have become harmful. For example, should we eliminate harmful species if that decreases biodiversity or put up dams to control flooding and generate electricity if that endangers the migration of salmon?[17] Should we drain malarial swamps and mitigate forest fire danger?

As we saw above, those who charge Christian theology with the responsibility for ecological problems usually attack the "dominion theology" of Genesis 1:28: "God blessed them [humans—male and female, vs. 27] and said to them, 'Be fruitful and increase in number; fill the earth and subdue it. Rule over the fish of the sea and the birds of the air and over every living creature that moves on the ground.'"

The most controversial concepts in this verse are expressed in the words "subdue" and "rule." In Hebrew, these are *kabash* and *radah*, respectively. To understand properly what Genesis 1:28 is saying, we must understand these Hebrew words. *Kabash* (the verb and its derivatives) occurs fifteen times in the Hebrew Bible. The Hebrew meaning can be expressed as "to subdue, bring into bondage."[18] The word is inescapably a harsh one. Typical of its use in the Hebrew Bible is to describe the conquest of the Canaanites in Palestine (Numbers 32:20–22); of forced servitude (Nehemiah 5:5); and even of rape (Esther 7:8).

> Despite recent interpretations of Gen 1:28 which have tried to make "subdue" mean a responsibility for building up, it is obvious from an overall study of the word's usage that this is not so. *kabash* assumes that the party being subdued is hostile to the subduer, necessitating some sort of coercion if the subduing is to take place. . . . Therefore "subdue" in Gen 1:28 implies that creation will not do man's bidding gladly or easily and that man must now bring creation into submission by main strength.[19]

The second word, *radah*, is somewhat less harsh but still a strong word. Its literal meaning in Hebrew is strong as well ("Come, trample the grapes, for the winepress is full," Joel 4:13). The meaning was extended figuratively in Hebrew to "have dominion, rule, dominate."[20] The word occurs twenty-two times in the Hebrew Bible with this meaning, although "Generally *radâ* is limited to human rather than divine dominion," for which the much more common word *mashal*, "to rule," is generally used.[21]

What then is the import of Genesis 1:28, and the strong verbs in it, especially remembering that the command was given before the Fall and before a curse was pronounced on the ground? To answer this, we must take account for two considerations, one exegetical and the other theological. The exegetical consideration comes from the immediately preceding context.

> Then God said, "Let us make man in our image, in our likeness, and let them rule over [*radah*] the fish of the sea and the birds of the air, over the livestock, over all the earth, and over all the creatures that move along the ground." So God created man in his own image, in the image of God, he created him; male and female he created them. God blessed them and said to them, "Be fruitful and increase in number; fill the earth and subdue it. Rule over the fish of the sea and the birds of the air and over every living creature that moves on the ground". (Genesis 1:26–28)

So contextually, there is a strong connection between "the image of God" and having dominion. While exegetes and theologians have offered a wide array of explanations for the meaning of the *imago Dei*, at a minimum, the meaning must be in the same semantic range as the idea of ruling. God is Maker, and therefore Ruler of Creation. In making humans in his image, God conferred on humans the status of vice-regents over creation, and this concept of delegated rulership lies at the heart of human stewardship of nature. Of all created things, living and non-living, only humans are endowed with the *imago dei*, and only humans are said to be "blessed" with the responsibility of dominion.

Many biblical texts support this view, for example Psalm 115:16: "The highest heavens belong to the Lord, but the earth he has given to man." Stewardship entails dominion. But we may immediately question why God placed human beings in a position of dominion over the rest of creation. Why should humans not live in complete *shalom* in nature, as the prophet's vision of the peaceful future kingdom celebrates?[22] What in creation needed subduing, given that God's verdict on what he created was "very good" (Genesis 1:31)?

The question brings us to the second consideration, the theological point of view on stewardship as restoration, and it has two aspects. First, Genesis tells us that evil is already present actively, not merely potentially, in the world *prior* to the fall of humanity. The serpent is already in the garden; the verdict that creation is "very good" is now contested. Whatever an interpreter makes of the serpent in Genesis 3, it is indisputable that the text presents it as figurative for or an embodiment of Satan. So the first theological point is that nature must be subdued to eliminate natural evil.[23] God is never portrayed as being threatened by Satan, but rather than eliminate the source of evil outright, God created men and women who, by their obedience to his rule, by their faithful exercise of their duties as stewards, would prevail over evil (both moral and natural) on God's behalf. It may seem that this aspect of stewardship has more to do with the restoration and care of the soul than the restoration and care of creation. But we understand from Job, for instance, or from many stories in the Gospels, that Satan and his hordes are permitted by God to have limited negative effects on the natural world. So combating natural evil is a theological duty bound up in having dominion.

The second aspect of the answer is this. Although "very good," creation is not said to be perfect. Nature itself—indeed the very matter out of which it has been built—is constituted of structures which, in the language of chaos theory, exhibit sensitive dependence on initial conditions. The world seems to be such that a minor mutation in a benign bacterium may result in a deadly strain that strikes down many. Random forces acting over time place irresistible pressure on a fault line, and the resulting earthquake kills thousands. The "butterfly effect" changes weather patterns, bringing drought or deluge, tornados or hurricanes. Seemingly innocuous errors in DNA replication result in tragic birth defects. A world such as this needs subduing and restoring.

And. in fact, in just those countries where a genuine biblical dominion theology has informed culture, much good has been done that falls under the rubric of "restoration."[24] Yes, critics are correct in pointing out that abuses of nature have been initiated based on a dominion theology, but I would argue, with others, that these are in fact instances of dominion abused. The critics must admit that human actions have cured polio and smallpox and many other disabling diseases and conditions. Though earthquakes, like weather, cannot be precisely predicted, they both can be planned for, and their effects mitigated. Reforestation in Niger can halt the advancing Sahara sands, preserving homes and preventing famine; agricultural research can produce hybrid corn, multiplying yield and feeding millions. Genetic research can greatly increase our understanding of our bodies, opening powerful new approaches to the prevention, diagnosis, and treatment of diseases. These are aspects of a true exercise of dominion, and cases where subduing nature can only be regarded as good. And if these examples all seem to beg the question by illustrating utilitarian results measured in human terms, then consider the examples of breeding endangered species in captivity and reintroducing them into their native ecosystems; of

restoring polluted lakes and rivers and coastlines; of rescuing beached whales or oil-soaked waterfowl.

To sum up: Because the natural world in which God placed Adam and into which we are born contains natural evil, and is a world in which the goodness of nature often hangs in precarious balance, responsible exercise of dominion is a duty, not an option, for humans made in God's image—whether believers or not.

This aspect of stewardship—having dominion, seeking restoration, or, as Francis Schaeffer called it, "substantial healing"—is alone sufficient, in my view, to legitimate much of natural science, engineering and technology. We should respect professional scientists and engineers, or science and math teachers, and take opportunities as we have them to encourage young people who have promise in these areas to become outstanding scientists for the glory of God and for the good of the world he made.

The harshness of the Hebrew words is not to be avoided, for the nature of the tasks of subduing and exercising dominion is not soft or easy; yet the harshness does not militate against the positive results of the tasks when properly carried out by God's stewards fulfilling their duty.

This restorative aspect of stewardship is one that naturalistic ethics has difficulty justifying. The popular "man-on-the-trail" environmentalist is likely to regard any human intervention as unnatural. But there is something fundamentally flawed with a view of the place of humanity in nature, which regards all human activity as somehow unnatural, and all technology as an intrusion to be resisted. A house built for human habitation is no more unnatural than a termite's mud castle or a beaver's lodge. That our shelters sometimes acquire Olympian proportions of wasteful opulence or grow into impersonal steel and glass highrises, or degenerate into dehumanizing urban slums is deplorable but no more "unnatural" than a series of beaver dams turning a free-flowing stream into a marsh, with the result that the stream no longer can support native trout or even a beaver population. (The difference, of course, is that humans have moral responsibility for such consequences while beavers do not.)

There is a twistedness in humanity that causes us to deploy our dominion over nature with fierce and destructive delight, and the true character, limits and purpose of dominion are easily forgotten. Humanity has frequently set itself up as a rival to God, usurping God's sovereign reign and arrogating absolute dominion for itself, often creating what God could never accept, unleashing on creation what God despises.

But a truly theocentric understanding of stewardship will ask questions about the legitimate employment of technology. It will weigh projected profits and efficiency benefits against environmental costs. Consequently, a theocentric ethic will not stop with the dominion aspect of stewardship, for conservation and preservation are equally features of stewardship.[25]

In sum, restorative stewardship is a rich and productive occupation when done aright. As men and women responsibly fulfill the stewardship duty to have dominion and subdue the earth, much good and great good can result.

## 5. Stewardship as Conservation

The difference between conservation and preservation may be illustrated by the comparison of the philosophies of the National Forest Service and the National Park Service respectively. Conservation, as practiced by the NFS, allows for a wide range of human activities on NFS lands; hence the motto, "Land of many uses." Conservation is conceived as management. The NPS, on the other hand, conceives its task as preservation, that is, maintaining the National Parks in a state as close as possible to their state prior to the arrival of humans (or, at least, the kind driving recreational vehicles). Overall, though, it seems reasonable to all but the most virulent anti-development environmentalist that both conservation and preservation are desirable and necessary features of a coherent environmental ethic. How does a stewardship ethic ground these two?

The conservation model of stewardship is initially supported in Genesis 2:15: "The Lord God took the man and put him in the Garden of Eden to work it and take care of it." The words translated "work" and "take care" are, respectively, the Hebrew words *'abad* and *shamar*. *'abad* is a common Hebrew word, generally translated "to work, serve." In reference to things, it is generally followed by an accusative of the thing upon which labor is to be expended; in Genesis 2:15 the accusative of reference is a feminine pronoun, while the word for garden is masculine. The most likely antecedent for the pronoun, then, is not the garden itself but the feminine word *adamah*, translated "ground" or "earth," that Adam was expected to "work."[26] The point then is that this service was not restricted to the Garden of Eden but was extended to the earth as a whole.

There is another important concept involved as well, for not only is *'abad* the most common Hebrew word for serving, but it is also a priestly word, the most common word for worshiping. For the Hebrew, worship consisted in doing what God commanded. Hence Adam's task in the Garden of Eden was, first of all, religious, not horticultural. By working the land he was serving God.[27]

The second word, *shamar*, while indeed meaning "take care of," often has about it the idea of guarding or preserving.[28] (I will return to this word below when considering stewardship as preservation.) In both serving and preserving the garden, then, Adam was exercising his delegated stewardship over God's domain. As beings made in God's image and serving as God's stewards of creation, humans subdue and have dominion *just so far as they serve creation*. The stewardship task is theocentric: humans serve—and worship—God *by* caring for creation!

Serving overlaps with, and helps define, the meaning of subduing. Serving in this sense is properly seen as conserving. The duty of serving clearly involves work; the idea is not simply to let things go wild. Here's one significant difference between conservation and preservation. Conservation values farmlands and pastures as sources of food and green spaces and gardens as places of beauty. In no way does a stewardship ethic encourage letting all land revert to its natural state. We can and must use natural resources, but prudently, with an eye to sustainability. Thus the nation of Israel is commanded to give a sabbatical to their farmlands by letting the land lie fallow every seventh year (Leviticus 25:1–7), and their failure to do this is cited as one of the reasons for the Babylonian captivity: "The land enjoyed its sabbath rests; all the time of its desolation it rested, until the seventy years were completed" (2 Chronicles 36:21). Of course, the obedience of the sabbatical year command demonstrated deep trust in the Lord's provisions, but it also would have had a very healthy restorative effect on the marginal soil of much of Palestine.

Not only farmlands, but farm animals were incorporated in Old Testament stewardship, as demonstrated by the commands already cited: "Do not muzzle an ox while it is treading out the grain" (Deuteronomy 25:4), and "A righteous man cares for the needs of his animal" (Proverbs 12:10). Louis Regenstein recounts in great detail the many biblical injunctions which lead to a conservation ethic and to caring for animal life, and then goes on to trace these ideas in the saints and thinkers of both the Eastern and the Western church.[29]

Of course, stewardship as conservation presupposes that the steward knows what to conserve. Here again the theocentric nature of stewardship becomes apparent, for the steward must value what the Creator values. Only by focusing theocentrically rather than biocentrically or ecocentrically can stewards ensure that they value what the Creator values and thus practice proper conservation. Still, the point is clear. A theocentric environmental ethic will certainly be sufficient to ground and justify ethical claims regarding conservation practices, and it is under the concept of stewardship as conservation that we can best understand and work out the value and means of sustainability.

While I cannot say much more here, it is under the rubric of sustainability that much of our practical decision-making must be considered. Are we consuming at an unsustainable rate—both with respect to our income levels, and with respect to our use of renewable and nonrenewable resources? Is our lifestyle a zero-sum affair, where our consumption means someone else's scarcity, where our overuse of resources means we leave our progeny a

depleted planet? Such issues should, I believe, be much more at the forefront of our day-to-day thinking than they have been in the past, and the answers surely involve questions of ethics.

## 6. Stewardship as Preservation

Preservation generally is taken to mean setting aside lands where naturally-occurring processes are allowed to unfold naturally, where human intervention is minimal and non-permanent. It might seem difficult to demonstrate that a biblically-based theocentric environmental ethic could ground the value of preservation. Although there are biblical texts which support restoration and conservation, there are no texts which clearly support the conception of stewardship as preservation. However, two lines of evidence do point this way.

First, there is the meaning of that word in Genesis 2:15, *shamar*, "to exercise great care over, guard, preserve."[30] It is clear, in the context of Genesis 1–2, that *shamar* must be consistent with *kabash*, *radah* and *'abad*, so the resulting concept cannot be restricted to preservation as the sole duty ethical with respect to creation. Rather, *shamar* must be seen as implying the care and preservation of that in God's creation which is worth preserving. But this, of course, returns us to the question of valuation. We must assign value in theocentric, not instrumental, terms. How does God value nature *simpliciter*? Listen to, as God speaking to Job, from the whirlwind (Job 38–39, *passim*):

Where were you when I laid the earth's foundation? Tell me, if you understand.

Have you journeyed to the springs of the sea or walked in the recesses of the deep?

Have you entered the storehouses of the snow or seen the storehouses of the hail?

Do you hunt the prey for the lioness? Who provides food for the ravens?

Do you know when the mountain goats give birth?

Do you watch when the doe bears her fawn?

Who lets the wild donkey go free? Who untied his ropes?

I gave him the wasteland as his home, the salt flats as his habitat.

He laughs at the commotion in the town; he does not hear a driver's shout.

The wings of the ostrich flap joyfully, but they cannot compare with the pinions and feathers of the stork.

She lays her eggs on the ground and lets them warm in the sand,

　　unmindful that a foot may crush them, that some wild animal may trample them.

She treats her young harshly, as if they were not hers;

　　she cares not that her labor was in vain,

　　for God did not endow her with wisdom or give her a share of good sense.

Yet when she spreads her feathers to run, she laughs at horse and rider.

In these verses, God fairly exults in the wild nature he has created, so it seems reasonable to conclude that any human attempt to completely domesticate or destroy wild places and wild things would amount to sacrilege.

Of course, not all of nature can or should be preserved in its wild state. But since God values the wild, so should his stewards. Preservation, deriving from an understanding of the value God places on his creation and his command to humanity to guard or keep creation, will find in wildness both a reflection of God's nature and a tonic for the human soul. Those of us who venture into wild country, where "man himself is a visitor and does not remain,"[31] would surely recommend it to all who are able. We find tranquility, and our souls are refreshed in the serene beauty of Yosemite Valley or Crater Lake. Our problems shrink to size, as do our egos, as we struggle up the rocks of Mt. Whitney or the glaciers of

Mt. Rainier. We gain perspective on our limited powers as we feel the force of the waves under a sea kayak or surfboard and come to understand God's blessing of nature—his command to living things to be fruitful and multiply—in the fertility of Monterrey Bay or the Everglades.

It is within the aspect of stewardship as preservation that we can locate duties with respect to endangered species: Humans are to preserve the wild creation, which God values as the product of his creative activity, and this would include the rich diversity of species. We could also locate duties to preserve wilderness areas that reflect "raw nature" to protect wild and scenic rivers and coastlands, tall-grass prairies, and rain forests. Here too, we would find duties to guide us in regulating human activity to avoid practices and products deleterious to wild nature, such as certain methods of crude oil transportation which risk spillage or unmitigated production and release of greenhouse gasses that contribute to global warming. Yet such stewardship must be carefully reasoned and based on the best science available rather than an emotional response to headlines.

Admittedly, discerning a balance between preservation, conservation, and restoration is very often quite difficult. Knowledge of nature, foresight of consequences, and humility in intervening—that is, wisdom—is necessary. But it is clear that all three duties can be grounded in a stewardship ethic.

## 7. Conclusions

I have shown that the criticisms of Christianity by White, Toynbee, and others can be met from the perspective of a biblical theocentric environmental ethic. But I have attempted to show more: The central concept of stewardship is rooted in an understanding of God as Creator and Master of the world, and value in nature derives from the relationship God maintains with his creation. I suggested that understanding our role as stewards of creation helps clarify moral judgments concerning the environment and supports a consistent structure of rights and duties with respect to nature. Specifically, a theocentric environmental ethic seems quite well suited for grounding the three general aspects of environmental ethics: restoration, conservation and preservation.

I claim, then, that a theocentric environmental ethic is worthy of closer examination regardless of one's religious views. Interestingly, Lynn White concluded his AAAS address by saying, "Since the roots of our troubles are so largely religious, the remedy must also be religious, whether we call it that or not. We must rethink and refeel our nature and our destiny." (White 1967) The noted environmental ethicist J. Baird Callicot agrees:

> The Judeo-Christian Stewardship Environmental Ethic is especially elegant and powerful. It also exquisitely matches the requirements of conservation biology. The Judeo-Christian Stewardship Environmental Ethic confers objective value on nature in the clearest and most unambiguous of ways: by divine decree. (Callicott 1994)

It seems then that the shift in ethical thinking required to embrace adequate environmental ethics may be, at its root, essentially religious. A theocentric stewardship environmental ethic shows us the way.

**Funding:** This research received no external funding.

**Institutional Review Board Statement:** Not applicable.

**Informed Consent Statement:** Not applicable.

**Data Availability Statement:** Not applicable.

**Conflicts of Interest:** The author declares no conflict of interest.

## Notes

[1] (White 1967). It has been enormously influential and reprinted in many anthologies.

[2] A convenient date for the birth of an organized environmental movement is the publication of Rachel Carson's *Silent Spring* in 1962.

3　That these are indeed the right issues on which to focus is illustrated by one of the most popular textbooks on the subject (Light and Rolston 2003). Part VI is titled "Focusing on Central Issues: Sustaining, Restoring, Preserving Nature." See also (Bassham 2020).

4　(James 1896); emphasis mine.

5　Illustrated, for example, by Beisner (1990). Beisner dismisses most environmental concerns as "ideologically influenced hysteria." Beisner is a founder of and the national spokesman for the Cornwall Alliance for the Stewardship of Creation; see www.cornwallalliance.org (accessed on 7 March 2023).

6　A *biocentric* ethic was laid out, for example, by Singer (1979). See also Taylor (1986).

7　An *ecocentric* environmental ethic is developed by Leopold ([1949] 1987) and by Rolston (1989).

8　Roots of the *resacralizing* approach are in (Lovelock 1979; McFague 1987; Naess 1986).

9　Lovelock (1979, p. vii); emphasis mine.

10　Among the many titles, see (Bouma-Prediger 2001; Hall 1990), hereafter *Steward*; (Nash 1993; Santmire 1985; Wilkinson 1990). See also "An Evangelical Declaration on the Care of Creation," originally signed by over one hundred evangelical leaders in 1994, available at http://www.creationcare.org/resources/declaration.php (accessed on 7 March 2023).

11　Like Nash, Paul Santmire clearly sees both sides, and his work is therefore somewhat richer than much writing from a Christian standpoint. Yet, as the subtitle to his book ("The Ambiguous Ecological Promise of Christian Theology") suggests, Santmire sees the subdue/have dominion theme as negative, in contrast to my more positive interpretation explained below.

12　An alternative to this deontological account is a virtue ethics approach. See Bouma-Prediger (2020). The appendix to this book is an extensive bibliography of works applying Christian virtue ethics to creation care. Of course, seeing and doing one's duty is a virtue.

13　See, for example, St. Thomas Aquinas, *Summa Contra Gentiles*, I.8.

14　Cited in Wilkinson (1990, p. 319); my emphasis.

15　See (Rolston n.d.) for an alternative approach.

16　(Blocher 1979); my translation.

17　The notion of restoration is contentious in environmental ethics. See (Elliot 1982; Katz 1992). My understanding of 'restoration' is somewhat different than that of Elliot or Katz.

18　(Brown et al. 1907), s.v.; hereafter cited as BDB.

19　(Harris et al. 1980), s.v. *kbš*; hereafter TWOT.

20　BDB, s.v. *rdh*.

21　TWOT, vol. 2, s.v. *rdh*.

22　"The wolf will live with the lamb, the leopard will lie down with the goat, the calf and the lion and the yearling together; and a little child will lead them. The cow will feed with the bear, their young will lie down together, and the lion will eat straw like the ox. The infant will play near the hole of the cobra, and the young child put his hand into the viper's nest. They will neither harm nor destroy on all my holy mountain" (Isaiah 11:6–8).

23　On natural evil, see DeWeese (2013).

24　See, for example, Stark (2003), who notes that both Whitehead and Oppenheimer are among those modern scientists who have emphasized, regardless of their personal religious beliefs, that modern science could only be born in a historic Christian cultural consensus with its belief that a reasonable God created the universe outside himself, and its principles therefore could be discovered by reason.

25　Santmire sees dominion not as restoration but as indifference to nature, and the eschatological posture of much of Christianity as deeply and emotionally opposed to devoting much effort to saving this world, which will be destroyed anyway as preparation for the new heavens and the new earth. Hence his subtitle, "The Ambiguous Ecological Promise of Christian Theology." His sociological observations about Christianity may, unfortunately, be justified, but I do not feel that his exegetical conclusions are adequate. I believe that restoration, conservation and preservation are equally a part of the larger stewardship model.

26　(Hamilton 1990), s.v. 2:15. Although not universally accepted as a correct interpretation, I am reading Genesis 2 as an amplification of Genesis 1. See Hamilton (1990, pp. 150–52).

27　(Cassuto 1961), s.v. 2:15.

28　Interestingly, in the Babylonian Creation Epic, commonly called the *Enuma elish* (after its first two words), the two concepts of serving and guarding are closely linked to the creation of man. See Pritchard (1968, Tablet VI).

29　(Regenstein 1991), passim.

30　TWOT, vol. 2, s.v. *šmr*.

31　The phrase is from the definition of 'wilderness' in the *Wilderness Act* of 1964.

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
