# Peer review of "A Theocentric Environmental Ethic"

_religions, doi:10.3390/rel14070913_

Round 1
Reviewer 1 Report
I’m not sure I understand the objections to the ecocentric ethic. What do natural, wild fluctuations in Earth’s natural history have to do with whether things in nature are intrinsically valuable? Humans might be intrinsically valuable even if we’re wiped out by a natural disaster. Unless you’re assuming that God is behind the natural disaster and so wouldn’t wipe us out by one if we’re intrinsically valuable. Also, even if things in nature are intrinsically valuable, why can’t God also use those things? Perhaps God cannot use things in nature as a mere means if He is obligated to respect their value (but does God even have obligations?). But this doesn’t mean He cannot use things as a means. Help me understand. At the very least, even if I disagree, I would like to understand just what the objection is. Also, and perhaps it’s because I don’t understand, how do we get to the conclusion that the creature’s value is prioritized above the Creator’s?
Re: duties to non-human species. Doesn’t the fact that certain animals are intrinsically valuable (at least those that are conscious) give us a duty to treat them in a certain way? It seems that I have a duty to not torture an animal for fun, not simply because I have a duty to steward something God owns, and not simply because of the psychological impact this would have on me. It’s the sort of thing that I ought not just treat any old way. I’m not saying you have to change anything in the article in light of this. I’m just offering some reflections.
I agree with the theological/philosophical position taken in the paper. I think it is argued well and does reflect a biblical perspective, especially compared to the other positions mentioned in the paper. But note: I am not in a position to analyze the claims made about various Hebrew terms. Perhaps another reviewer can.
Author Response
Thank you for your comments.
(1) As to the relevance of natural fluctations--that is directed to the ecocentric value of preserving ecosystemic homeostasis. The section questions the intrinsic value of nature claimed to be supported by the ecocentric view.
(2) I l leave open the question of whether (some) species have intrinsic value, but question whether, even if they do, such nonmoral entities can impose duties on moral subjects (humans). I try to show that a more satisfying view is to see duties to God with respect to natural entities. I have not argued for a psychological ground of duties.
Reviewer 2 Report
"A Theocentric Environmental Ethic" offers an unusually rich--albeit somewhat dated--engagement with the literature on Christian environmental ethics and for that reason alone stands to make a contribution to scholarship. However, its engagement with that literature--whether critical or constructive--is not original. Accordingly, the article would seem more an encyclopedia / handbook / companion entry than a journal article but for the fact that it offers an ambitious constructive thesis. Nevertheless, that thesis is never fully specified and fluctuates wildly, from the claim that "Christians are uniquely able to offer a framework for environmental ethics" (39) to "only an ethic of creation care which is theocentric at its core ... [is] biblically faithful" (142-144) to "criticisms of Christianity by White, Toynbee, and others can be more than met" (636-637) and finally "a theocentric environmental ethic seems quite nicely--perhaps uniquely--situated for grounding the three general aspects of environmental ethics which do not rest easily together in other systems: restoration, conservation, and preservation [and thus] theocentric environmental ethics is worthy of closer examination and application by religious believers and nonbelievers alike" (643-648).
Admittedly, the first and last of these claims are similar, and the author evidently takes the second claim--that "only an ethic of creation care which is theocentric at its core ... [is] biblically faithful" to support them. Yet if so, the basic argument of the paper is obscure and debilitatingly incomplete. While the author announces their commitment to Christianity (44-45), the author never explicitly states the purposes of their argument or that they will take Christian scripture as normative. However, the paper apparently take Christian scripture as normative and so, at a minimum, should make this claim explicit. But even once explicitly made, the paper's apparently central argument, that a theocentric environmental ethic "is uniquely able to offer a framework for environmental ethics ... [that] is worthy of closer examination and application by religious believers and nonbelievers alike" is overstated and unconvincing. Of course, other perspectives able able to offer a framework for environmental ethics. And why should nonbelievers, who presumably do not take Christian scripture as normative, find theocentric environmental ethics worthy of closer examination if the paper's argument for the cogency of theocentric environmental ethics is based on the authority of Christian scripture?
In short, the paper seems methodologically conflicted between attempting to show that a theocentric environmental ethic is supported by scripture and attempting to argue that a theocentric environmental ethic is the superior framework for environmental ethics, however it is supported. Of course, it is possible to have and accomplish both aims. However, establishing the first aim does not establish the second without an argument for the authority of Christian scripture in environmental ethics that the paper does not provide.
Admittedly, the second section of the paper, "Contemporary Perspectives: A Brief Summary" is just as described: a cursory review and dismissal of four prominent, broad alternatives in environmental ethics. This helpfully contextualizes the author's constructive proposal of a theocentric environmental ethic and thus this section should remain in the article. However, it is insufficient to demonstrate the superiority of theocentric environmental ethics and hence here again the paper's evident thesis--that a theocentric environmental ethic is the best environmental ethic whether or not one is a theist--remains unwarranted.
In addition to this fundamental, methodological concern, I mention four subordinate matters for the author to consider:
1) is it cogent to claim "all aspects of the natural world have intrinsic value" (305) and to deny that any aspects of the natural world possess rights "(349-351)?
2) the treatment of restoration (section 4) is underdeveloped. The term is never defined, only illustrated (e.g. 359-362), and given the author's subsequent assertion that natural evil is a distortion of God's creation (427-431), the author claims the duty of restoration entails prevailing over (434) and combatting (439) natural evil. But how could human beings prevail over natural evil? And how is prevailing over or combatting natural evil consistent with the duty to preserve nature that the author subsequently affirms? Specifically, the author contends "preservation, deriving from an understanding of the value God places on his creation and his command to humanity to guard or keep creation will find in wildness both a reflection of God's nature and a tonic for the human soul" (614-617). But nature and wildness involve natural evil. Accordingly, as the author describes them, restoration and preservation appear incompatible.
3) while the article identifies several fruitful aspects of envisioning human beings as environmental stewards and imago dei, it does not address a problem with these ideas, namely that the steward "manages the estate in the manner the master would want it managed were he present" (293-294). Likewise, the author cites Old Testament scholarship to claim that the imago dei and dominion theology of Genesis 1 are rooted in "the 'royal ideology' of the ancient near East, where a statue or viceroy functioned as the symbolic image or representative of the ruler's authority over a territory or people in the ruler's absence" (410-413). Can Christians who accord authority to Scripture accept the implication of stewardship and dominion exercised by those bearing the imago dei that God is absent from God's creation? If not, can they affirm the notion of human beings as environmental stewards or imago dei so understood? Of course, no metaphor / symbol is perfect and thus this objection need not prove decisive. Nevertheless, the article would be stronger if it acknowledged this liability of stewardship and imago dei / dominion theology, and better still, if it mitigated this liability.
4) given the author's already noted assumption of Scriptural authority, the author's claim that "Only by focusing theocentrically rather than biocentrically or ecocentrically can stewards ensure that they value what the Creator values and thus practice proper conservation" (559-561) begs the question, since it presupposes rather than demonstrates that proper conservation is defined by what the Creator values.
Reviewer 3 Report
I have uploaded my annotated copy of the paper. My remarks are in red for visibility.
I found several important gaps in the argument; if author would address them this could be a helpful contribution.

Reviewer 4 Report
This paper presents an important contribution to an increasingly contentious subject. It has thoughtful and original content. I recommend publishing this paper; however, there are several issues to be addressed, ranging from formatting problems to content and structure problems, as I discuss in my line-by-line comments below:
37-8: Any examples that could be cited? I imagine many Jews and Christians might find the thesis agreeable. In fact, the paper later will confirm this (e.g., 84). Perhaps you could switch this to a more objective assessment of the thesis you are criticizing, not dependent on the reception of the thesis.
41-43: I was expecting a stronger thesis, such as that the “rape of nature” is, in fact, the natural outcome of materialism or religious liberalism, and that any criticism of such a rape borrows from the Bible.
44: “I need to acknowledge where I stand.” I don’t understand why that is true?
44-54: In addition to the personal beliefs, this diversion into the contentious topic of global warming seems inappropriate, irrelevant, distracting, and raises the question of whether or not this is a scholarly paper and worth the reader’s investment of time and energy. I certainly would not read further at this point if I wasn’t a reviewer.
57: “This will serve …” What will serve? There is no subject.
58: “I’ll”. Avoid contractions in a journal paper.
65: “utilitarian.” I did not detect this in the following quotes.
88f: Again, I am perplexed by the insertion of personal beliefs. Not only that, but it muddies the discussion that follows. Legitimate for whom? Christians? What hierarchy of values? “Our primary duty …” Who does this refer to? Christians? Is this a paper for Christians? “I’ll let the positive case for stewardship of creation speak for itself.” Aside from the contraction, this seems to be close the thesis of this paper. But if so, you certainly would not be so indifferent. If not, then this is confusing.
97: “also seem to me to be flawed. ”But you did not show the first one to be flawed. In fact, aside from when held by Christians, you did not even assert it was flawed.
116: “cogently.” Are you not criticizing this view? If so, referring to an advocate as cogent is confusing.
125: “than” should be “then.”
127: Argues this is problematic for Christians, but this section does not criticize the ecocentric ethic from a non Christian view, as did the previous section. So the paper is inconsistent regarding audience.
136-7: “childish.” Confusing. How/why is “Mother Earth” childish?
138-9: Now here is a utilitarian argument. But you did not identify it as such, whereas you did back at line 65, where it was not clear.
140: You need to spell out how genocide is problematic; otherwise, this is not clear.
142: You need to work harder at showing inadequacy, and summarize your key points here. I found inadequacy only in the Biocentric critique, based on a logical fallacy, described in the final sentence of Section 2.2. For the minimalist and ecocentric, you suggested inadequacy, but merely for Christians. For deep ecology, you suggested inadequacy based on a vague reference to genocide, but you did not actually provide the argument; you merely stated that it should be obvious. The paper, as presented, hinges on your making this inadequacy case for all four views, and you have merely touched lightly on this.
142-44: “only an ethic of creation care which is theocentric.” The word “only” makes this a high claim. Presumably you will make this argument in the remainder of the paper. Why not state that here to help the reader know where we are? As it stands, this is merely a bare assertion.
228-9: Perhaps clarify that you are referring to mountains, for readers less familiar with this geography.
253: Again, I would omit the personal ax to grind. Make the case, without resorting to “I would insist …”, which diminishes the paper, rendering it as partisanship.
336: This section needs a conclusion, rather than ending on a quote. What’s the conclusion?
350: Avoid declaring what side you are on.
410-418: Is this a quote? If so, should be formatted as such.
492: Why “admittedly”?
512: What is a Winnebago?
571: “that” should be “than”
637-8: Sentence does not make sense.
Please see above comments on use of contractions.
Round 2
Reviewer 3 Report
I notice overlap between this article and the PhD thesis by Christopher Cone, Redacted Dominionism: An Evangelical, Environmentally Sympathetic Reading of the Early Genesis Narrative (U of North TX PhD in Philosophy), August 2011. Is this the same author? If the author here thinks things are more fully developed in the thesis, he might refer the reader to it; otherwise many statements in the paper sound like assertions without proper support.
While the author has taken on board a few of the previous comments, I still think the ones he/she ignored could make the paper’s argument more compelling. I will leave it up to the author whether or not to improve the paper further—my comments are meant to help.